# Effect of UV-Irradiation and ZnO Nanoparticles on Nonlinear Optical Response of Specific Photochromic Polymers

**DOI:** 10.3390/nano11020492

**Published:** 2021-02-16

**Authors:** Karolina Waszkowska, Tarek Chtouki, Oksana Krupka, Vitaliy Smokal, Viviana Figà, Bouchta Sahraoui

**Affiliations:** 1Laboratory MOLTECH-Anjou, University of Angers, CNRS UMR 6200, 2 Bd Lavoisier, 49045 Angers CEDEX, France; 2Materials and Subatomic Physics Laboratory, Superior School of Technology, Ibn-Tofail University, Kenitra 14000, Morocco; tarek.chtouki@uit.ac.ma; 3Faculty of Chemistry, Taras Shevchenko National University of Kyiv, 60 Volodymyrska, 01033 Kyiv, Ukraine; oksana_krupka@yahoo.com (O.K.); vitaliismokal@gmail.com (V.S.); 4Euro-Mediterranean Institute of Science and Technology (IEMEST), via Michele Miraglia 20, 90100 Palermo, Italy; viviana_fg@yahoo.it

**Keywords:** photochromic polymers, methacrylic styrylquinoline polymers, ZnO nanoparticles, UV irradiation, nonlinear optics, second harmonic generation, third harmonic generation, nonlinear susceptibility

## Abstract

A series of methacrylic styrylquinoline polymers have been synthesized and characterized by spectroscopic and nonlinear optical (NLO) investigations. The NLO properties of studied polymer compounds in the form of thin films prepared by a spin coating method have been investigated by means of second and third harmonic generation via Maker fringe setup with a laser source at 1064 nm and a pulse duration of 30 ps. The results show strong second harmonic signal dependence on polarization configurations. This second harmonic generation (SHG) response was enhanced by UV-irradiation at 366 nm and doping by ZnO nanoparticles (NPs) (100 nm), while the opposite effect was achieved for a third harmonic generation experiment. Thus, values of second and third order nonlinear susceptibilities were determined by theoretical calculations based on comparative models. The remarkable NLO results presented in this paper expose potential optoelectronic and photonic applications.

## 1. Introduction

In recent years, there has been a growing research interest in developing materials with potential for application in photonics, optoelectronics, optical signal processing, and optical computing [1,2,3,4,5]. However, the choice of materials for nonlinear optical (NLO) investigations depends not only on their optical characteristics. The basic properties that are required for advanced inorganic and organic nonlinear optical materials include high NLO chromophore density (polymers), display large optical nonlinearity, low optical losses, and ultrafast response time, as well as transparency in the range of the used wavelengths, resistance to high laser light intensity, and stable optical, thermal, and mechanical parameters [6,7,8,9].

Nonlinear optical polymers have significant perspectives for applications due to certain advantages over inorganic ones [10,11]. The NLO active moieties can be either doped into the polymer matrices or covalently attached to the polymer [12,13]. Polymers with chromophore units in the main chain or in the side chain usually result in more stable systems [14] with increased density of chromophores and an enhanced nonlinear optical response. The strong nonlinear response comes from the strong charge transfer taking place within these units [15].

Among all types of polymers with chromophore fragments, polymers with styrylquinoline moiety have been chosen due to several advantages: styrylquinoline methacrylic polymer has a chromophore with two functional groups, a central (ethylenic) double bond, and an endocyclic (quinoline) nitrogen atom, and results in photoisomerization of the ethylenic group, which allows switching between two isomeric states, known as trans (E) and cis (Z), excluding any side photochemical reactions. Both isomeric states are thermally stable. Protonation of the nitrogen atom allows shifting of the absorption spectrum. The photoisomerization and protonation process are reversible. 

It should be noted that, in addition to the properties already listed, in our previous research [16,17,18], significant perspectives of methacrylic polymers were reported with chromophores in the side chain for applications, such as optical switches, optical data storage, and holography. The method of ZnO/polymethacrylate-based nanocomposite film preparation and the investigation of optical absorption, photoluminescence, and nonlinear optical properties were observed [19,20,21,22]. However, we focus on NLO properties of new styrylquinoline polymers thin films with different substituents in the chromophore moiety and the influence of ZnO nanoparticles (NPs) of a nanocomposite and UV irradiation on the nonlinear optical response of the materials.

## 2. Materials and Methods

### 2.1. Methodology

2,2′-Azobis(isobutyronitrile) (AIBN) was recrystallized twice from absolute methanol. Methacrylic chloride was vacuum distilled, immediately before use. Methylmethacrylate (MMA) was washed with aq NaOH to remove inhibitors and dried with CaCl_2_ under nitrogen at reduced pressure. Standard distillation procedures were performed for triethylamine, N-N-dimethylformamide (DMF), and tetrahydrofuran (THF) prior to use. All other reagents, solvents, and nano-powder ZnO (<100 nm in size) were commercially available and used as received.

NMR spectroscopy: ^1^H NMR (400 MHz) spectra were recorded on a Mercury (Varian Inc., Palo Alto, CA, USA) spectrometer in DMSO-d6 at room temperature. Chemical shifts are given in ppm from tetramethylsilane. 

UV-VIS measurements were performed at room temperature either in solutions in a quartz liquid cell, or as thin films deposited on glass substrates with a Shimadzu UV-1800 spectrometer (Shimadzu, Kyoto, Japan).

Differential Scanning Calorimetry (DSC): A Q20 model DSC (TA Instruments, New Castle, PA, USA), with a continuous N_2_ purge, was used to determine the glass and phase transition temperatures (T_g_) of all polymers.

Gel Permeation Chromatography (GPC): The molecular weights of all polymers were evaluated with Spectra SYSTEM RI-150 and Spectra SYSTEM UV2000 detectors (Thermo Scientific Corp., Waltham, MA, USA). Spectra SYSTEM AS1000 autosampler come with a guard column (Polymer Laboratories, PL gel 5 μm Guard, 50 mm × 7.5 mm) followed by two columns (Polymer Laboratories, 2 PL gel 5 μm MIXED-D columns, 2 mm × 300 mm × 7.5 mm). The eluent used is THF at a flow rate of 1 mL/min at 35 °C. Polystyrene standards (580–4.83 g/mol × 103 g/mol) were used for calibration.

### 2.2. Synthesis and Thin Films Preparation

#### 2.2.1. 2-(2-Biphenyl-4-ethenyl)quinolin-8-yl propionate (1p)

A round bottom flask was charged with a mixture of 8-hydroxy-2-methylquinoline (3 g, 18.8 mmol), biphenyl-4-carboxaldehyde (6.5 g, 36 mmol), and propionic anhydride (PA) (25 mL). The mixture was then moved to an oil-bath and refluxed with heat at 140 °C for 10 h (monitoring by thin layer chromatography (TLC)). For the cooled reaction mixture, ice-cold water was added. The precipitated product, which appeared upon dilution with an ice/water mixture, was filtered off, washed several times with water, dried and crystallized from ethanol, yielded 60%, Mp 165 °C. ^1^H NMR (400 Hz, DMSO-d6), δ, ppm: 1.38 (t, 3H, –CH_3_), 2.83 (m, 2H, –CH_2_–), 7.31–7.51 (m, 4H, Het), (m, 1H, =CH–), (m, 4H, Ar-H), 7.62–7.81 (m, 1H, =CH–), (m, 5H, Ar–H), 8.29 (d, 1H, Het).

#### 2.2.2. 2-(2-Biphenyl-4-ethenyl)quinolin-8-ol (1 h)

The solution of 1p (3 g, 8 mmol) in ethanol (80 mL) and concentrated hydrochloric acid (20 mL) was refluxed for 2 h. The precipitate was filtered and washed thoroughly with water. It was subsequently dissolved in ethanol (20 mL) and triethylamine (120 mL) was added and stirred at room temperature for 1 h. The ice-cool water was added and the precipitate was filtered, washed with water, and dried to afford 1 h as a dark yellow powder. It was purified by recrystallization from ethanol, yield 90%, Mp 178 °C. ^1^H NMR (400 Hz, DMSO-d6), δ, ppm: 7.03 (d, 1H, Het), 7.28 (d, 1H, Het), 7.33 (t, 1H, Het), 7.41–7.57 (m, 3H, Ar–H, =CH–), 7.71 (d, 1H, Ar–H), 7.78–7.87 (m, 4H, Ar–H, =CH–), 8.06 (s, 1H, Ar–H), 8.18–8.24 (m, 2H, Het), 9.25 (s, 1H, –OH).

#### 2.2.3. 2-(2-Naphtyl)ethenyl)quinolin-8-yl-propionate (2p)

The same procedure as for 1p was used with 2-naphthaldehyde. The solid residue was recrystallized from ethanol to give 2p yield 92%, Mp 155 °C. ^1^H NMR (400 Hz, DMSO-d6), δ, ppm: 1.42 (t, 3H, –CH_3_), 2.87 (m, 2H, –CH_2_–), 7.41 (d, 1H, Het), 7.48–7.51 (m, 1H, =CH–), (m, 2H, Ar–H), 7.77 (d, 1H, Het), 7.85–7.94 (m, 1H, =CH–), (m, 2H, Het), (m, 4 H, Ar–H), 8.03 (s, 1H, Ar–H), 8.32 (m, 1H, Het).

#### 2.2.4. 2-(2-Naphtyl)ethenyl)quinolin-8-ol (2h)

The same procedure as for 1 h was used. Yellow-orange powder of 2 h was obtained with a yield of 87%, Mp 170 °C. 1H NMR (400 Hz, DMSO-d6), δ, ppm: 7.04 (d, 1H, Het), 7.26 (d, 1H, Het), 7.33 (t, 1H, Het), 7.46–7.58 (m, 2H, Ar–H, =CH–), 7.71 (m, 1H, Ar–H), 7.83–7.88 (m, 5H, Ar–H), 8.06 (s, 1H, Ar-H), 8.18–8.24 (m, 2H, Het, =CH–), 9.20 (s, 1H, –OH).

#### 2.2.5. 2-[2-(4-Nitrophenyl)ethenyl]quinolin-4-yl-propanamide (3p)

The same procedure as for 1p and 2p was used with 4-nitrobenzaldehyde and 4-aminoquinaldine for obtaining 3p, Mp 168 °C. ^1^H NMR (400 Hz, DMSO-d6), δ, ppm: 1.2 (t, 3H, –CH_3_), 2.58 (m, 2H, –CH_2_–), 7.56 (t, 1H, Het), 7.64 (d, 1H, =CH–), 7.73 (t, 1H, Het), 7.84 (d, 1H, =CH–), 7.96–8.01 (m, 2H, Ar–H), (m, 1H, Het), 8.14 (d, 1H, Het), 8.24 (d, 2H, Ar–H), (s, 1H, Het), 10.61 (s, 1H, –NH).

#### 2.2.6. 2-[2-(4-Nitrophenyl)ethenyl]quinolin-4-amine (3h)

The same procedure as for 1h was used. The 3h yellow powder was obtained with a yield of 85%, Mp 249 °C. ^1^H NMR (400 Hz, DMSO-d6), δ, ppm: 6.69 (s, 2H, –NH_2_), 6.8 (s, 1H, Het), 7.32 (t, 1H, Het), 7.44 (d,1H, =CH–), 7.56 (t, 1H, Het), 7.71 (d, 1H, =CH–), 7.76 (d, 1H, Het) 7.87 (d, 2H, Ar-H), 8.07 (d, 1H, Het), 8.24 (d, 2H, Ar-H).

#### 2.2.7. 2-(2-Biphenyl-4-ethenyl)quinolin-8-yl-2-methylpropil-2-enoate (M1)

A solution of 1h (2 g, 6.1 mmol) and triethylamine (2 mL) was dissolved in THF (5 mL). The solution was kept in an ice bath for 10 min. A solution of distilled methacryloyl chloride (2 mL, 19.13 mmol) in THF (5 mL) was added slowly to the reaction mixture. After the addition of methacryloyl chloride, solution was stirred for 4 h in an ice bath and then poured into water. The light brown powder was collected by filtration, washed with water, and dried. The product was recrystallized from toluene, yield 50%, Mp 140 °C. ^1^H NMR (400 Hz, DMSO-d6), δ, ppm: 2.21 (s, 3H, –CH_3_), 5.98 (s, 1H, =CH_2_), 6.5 (s, 1H, =CH_2_), 7.35–7.39 (m, 2H, Het, Ar–H), 7.43–7.47 (m, 3H Ar–H, =CH–), 7.53 (t, 1H, Het), 7.63–7.71 (m, 6H, Ar–H, Het, =CH–), 7.75–7.79 (m, 3H, Ar–H), 8.35 (d, 1H, Het).

#### 2.2.8. 2-[2-(4-Nitrophenyl)ethenyl]quinolin-4-yl-2-methyl-2-propenamide (M2)

The yellow solid was received, yield 80%, Mp 177 °C. ^1^H NMR (400 Hz, DMSO-d6), δ, ppm: 2.20 (s, 3H, –CH_3_), 5.6 (s, 1H, =CH_2_), 6.0 (s, 1H, =CH_2_), 7.5 (t, 1H, Het), 7.6 (d, 1H, =CH–), 7.53 (t, 1H, Het), 7.6 (d, 1H, =CH–), 7.69 (t, 1H, Ar–H), 7.8–7.92 (m, 3H, =CH–, Ar–H), 8.12–8.22 (m, 3H, Het, Ar–H), 10.01 (s, 1H, –NH).

#### 2.2.9. 2-(2-Naphtyl)ethenyl)quinolin-8-yl-2-methylpropil-2-enoate (M3)

The yellow solid of M3 was obtained, yield 60%, Mp 115 °C. ^1^H NMR (400 Hz, DMSO-d6), δ, ppm: 2.24 (s, 3H, –CH_3_), 5.99 (s, 1H, =CH_2_), 6.5 (s, 1H, =CH_2_), 7.43–7.52 (m, 5H, Het, Ar–H, =CH–), 7.76–7.96 (m, 8H Ar–H, =CH–, Het), 8.27 (d, 1H, Het).

Polymerization: The styrylquinoline copolymers were synthesized by free-radical polymerization. The polymerization was carried out in 10 wt% DMF solution of M1 or M2, M3 (0.005 mol), and MMA (0.015 mol). The polymerization was conducted using AIBN as a free radical initiator (1 wt% of monomer) at 80 °C for 24 h in an argon atmosphere. Previously, the initial mixture was degassed with repeated freeze-pump-taw cycles. Then, the reaction mixture was poured into methanol. This procedure was repeated several times to ensure removal of unreacted compounds and, finally, the precipitate was separated from methyl alcohol and dried under vacuum at 50 °C overnight. The copolymerization ratios were calculated on the basis of the integrated peak areas of ^1^H nuclear magnetic resonance (NMR) spectra (Table 1).

Thin films preparation: The ZnO nanopowder from Sigma-Aldrich (<100 nm in size) have been used for preparation of the nanocomposite film P1-NPs. The glass substrates were carefully cleaned in a commercial surfactant using ultrasonication and washed several times in deionized water. The cleaning procedure was ended by annealing in a 200 °C oven for 60 min. Then spin coating technique was used to fabricate nanocomposite film with ZnO nanocrystals dispersed in P1. The solution of 1,2,2-trichloroethane containing P1 100 g/L and ZnO was coated on BK7 glass slides. The concentration of ZnO to P1 was 10% wt corresponding ZnO nanoparticles. The thin film was prepared by mixture depositions using a spin coater (Spin200i, POLOS, Putten, Netherlands) at a spin rate of 2000 rpm for 60 s. The thicknesses of prepared thin films were measured by using a profilometer (Dektak 6M, Veeco, Tucson, AZ, USA) and presented in Table 2.

The derivatives of 8-hydroxy-2methylquinaline and 4-amino-2-methylquinaline have been synthesized by a reaction of condensation with propionic anhydride (PA) and, consequently, benzaldehydes. Methacrylic monomers M1, M3, and M2 were synthesized by a reaction of the alcohol or amino derivatives, respectively, with methacryloyl chloride in the presence of triethylamine. The corresponding copolymers P1, P2, and P3 were obtained by radical polymerization of these monomers with MMA using AIBN as a radical initiator (Figure 1).

The structures of the copolymers were confirmed by ^1^H NMR spectra and a reasonable accordance was found between the observed n/m values in the polymers in Table 1. Nevertheless, as expected from higher polymerization ability of methacrylic esters with a styrylquinoline unit [17], the MMA motifs are generally found to lack in the final composition of the polymers P1 and P3. Glass transition temperatures (T_g_) are collected in Table 1. As expected, introduction of conformationnally flexible –O– as linkers of the styrylquinoline chromophore leads to a decrease of T_g_ (P1 and P3 vs. P2).

Finally, the copolymers molecular weights are in the limit of 15,000–27,900 g/mol and polydispersity indices are 1.84–2 as determined by GPC (Table 1). The thin film containing the P1 polymer and ZnO nanoparticles was spin-deposited on a glass substrate to investigate the influence of NPs on nonlinear optical features of the nanocomposite due to the fact that the interfacial interaction between ZnO NPs and the polymer matrix plays an important role in improving the final optical properties of material.

### 2.3. Second and Third Harmonic Generation Experiment

The second harmonic generation intensity and the formation of characteristic fringes were first observed experimentally in 1962 by P. D. Maker [23]. The experiment consisted of focusing a ruby laser beam on the surface of a quartz crystal, and then measuring how the generated second harmonic signal intensity changed as the crystal was rotated, thus, changing the effective optical path length through the crystal. In our research work, second and third harmonic generation (SHG and THG) measurements were carried out using an experimental setup presented in Figure 2. The Nd:YAG laser beam (PL2250, Ekspla, Vilnus, Lithuania) with a wavelength of 1064 nm, a power of 90 µJ, a pulse duration of 30 ps, and a repetition rate of 10 Hz, passing through a series of optical elements, was focused on the sample through the lens at a focal distance of 25 cm, and then the sample was rotated from −75 degrees to the position perpendicular to the laser beam and then up to 75 degrees. The SHG and THG measurements were performed in two configurations: by changing the angle on the half-wave plate (λ/2), we are able to obtain an s-polarized and p-polarized laser beam, s and p. The separated signal, by applying an interference filter (355 nm for THG and 532 nm for SHG, respectively), was collected by a photomultiplier tube and then processed into graphical data.

## 3. Results and Discussion

### 3.1. Spectroscopic Studies

In the spectrum of absorption, the THF solutions of P1, P2, and P3 have intensive long-wave bands with a maximum at 348 nm, 356 nm, and 348 nm, respectively, and absorption band with its maximum at about 300 nm, 311 nm, and 278 nm. The electron-donor and electron-acceptor substituents in the copolymers P2 is increasing the charge transfer character of the π–π* transition and consequently shifting the π–π* band to the red (Figure 3 and Figure 4). Thereby, P2 exhibits a maximum absorption at the longest wavelength (356 nm) among the investigated compounds that can be explained by the highest charge-transfer interaction occurring between the electron-donor group (amino) and the electron-acceptor group (nitro). 

UV-vis spectra of studied polymer thin films are presented in Figure 4. It was observed that, at a fundamental laser wavelength of 1064 nm, absorbance is negligible. However, the same situation is at wavelength 532 nm, which corresponds to a generated second harmonic, which means, at these wavelengths, there is no contribution of absorption to the NLO response. Nevertheless, at a wavelength of 355 nm corresponding to a generated third harmonic, absorbance is significant, which means that, in calculations of third-order nonlinear optical susceptibility, it is necessary to take absorption contribution into account. Determined values of an absorption coefficient (α) for wavelength 355 nm are given in Table 2.

The decreases of absorption intensity at 348 nm and 300 nm and increases of optical density at 238 nm with an isosbestic point at 280 nm were observed (Figure 5) during the P1 solution irradiation process with an emission wavelength spectrum at 365 nm (UV lamp 25 W). The same behavior was observed for P2 and for P3 with an appearance of an isosbestic point at 296 nm and 249 nm, respectively. It can be explained by trans-cis isomerization in P1, P2, and P3 molecules. For the SHG and THG experiment, polymer thin films of P1, P2, and P3 were irradiated for 30 min at 20 °C to produce the corresponding cis isomer in the side chain of polymers at 365 nm (UV lamp of 25 W).

The common photochemical properties of diarylethylene compounds are their possibility for photoisomerization through rotation around the C–C bond. As an example, Figure 4 shows changes in the absorption spectra of P1 due to optically induced transition to the cis-isomer. It is impossible to convert completely one isomer into another photochemically because most of diarylethylenes isomerizes reversibly by an adiabatic mechanism [24]. During irradiation, the photo-stationary equilibrium can be reached and the concentrations trans-isomers and cis-isomers in the mixture depend on the irradiation wavelength [25]. Commonly, the transformations are always accompanied by changes in the geometry of the molecule from the planar to the non-planar. There are changes in the dipole moment as well.

### 3.2. NLO Properties

Second-order nonlinear optical susceptibility was calculated by using the Lee model [26].
(1)$$\chi^{\left(2\right)}=\chi_{Quartz}^{\left(2\right)}\left(\frac{2}{\pi }\right)\left(\frac{L_{Quartz}^{coh}}{d}\right)\sqrt{\frac{I^{2\omega }}{I_{Quartz}^{2\omega }}}$$
where $\chi_{Quartz}^{\left(2\right)}=1\cdot 10^{-12}\;m\cdot V^{-1}$ [27], $L_{Quartz}^{coh}=21\;\mu m$ is the coherence length of reference material, *d*—sample thickness, and $I^{2\omega }$ and $I_{Quartz}^{2\omega }$ are SHG intensities of thin film and reference material, respectively. However, third-order nonlinear optical susceptibility was calculated by using the Kubodera-Kobayashi model [28].
(2)$$\chi^{\left(3\right)}=\chi_{Silica}^{\left(3\right)}\left(\frac{2}{\pi }\right)\left(\frac{L_{Silica}^{coh}}{d}\right)\left(\frac{\frac{\alpha d}{2}}{1-exp\left(-\frac{\alpha d}{2}\right)}\right)\sqrt{\frac{I^{3\omega }}{I_{Silica}^{3\omega }}}$$
where: $\chi_{Silica}^{\left(3\right)}=2\cdot 10^{-22}m^{2}\cdot V^{-2}$ [29], $L_{Silica}^{coh}=6.7\;\mu m$ is the coherence length of reference material, *α*—linear absorption coefficient, *d*—sample thickness, and $I^{3\omega }$ and $I_{Silica}^{3\omega }$ are THG intensities of thin film and reference material, respectively. Errors of determined χ^(2)^ and χ^(3)^ values have been calculated by an exact differential function.

Second and third harmonic generation measurements were carried out by using an experimental setup presented in Figure 2. SHG and THG intensities as a function of the incident angle are presented in Figure 6 and Figure 7 and calculated values of second-order and third-order nonlinear optical susceptibilities given in Table 2. It was observed that, in case of THG, there is no dependence on polarization in the NLO response. We also noted that there are no significant discrepancies between the response-depending sample before and after UV irradiation. Admittedly, UV irradiation caused an NLO response to be slightly lower, but the χ^(3)^ values are not extremely varied. It was also noted that the addition of ZnO NPs influenced the second-order and third-order NLO response. Values of determined second and third order nonlinear susceptibilities are slightly lower. The highest third-order NLO response has been observed in the P1 sample. SHG measurements had to be carried out after a corona poling technique [30]. Therefore, second harmonic generation strongly depends on the symmetry of the molecule. Corona poling is the creation of macroscopic noncentrosymmetry by applying a high voltage (6 kV) that orients the molecules along the lines of the applied electric field. In addition, the sample must be heated to a temperature close to the glass transition temperature of the material. After 10 min, the system is cooled gradually to room temperature and after, as the system is turned off, the created orientation of the dipole moments remains suspended for a longer time. All studied thin films were poled at the same optimal conditions. The ordering is observed by the decrease of the thin film optical absorption at a normal incidence after corona poling [31]. Figure 6 shows the second harmonic generation signal of P1, P2, and P3 polymers after using the corona poling technique (CP, dashed line) and, after irradiation with ultraviolet light (UV + CP, solid line) and a sample doped with ZnO nanoparticles (NPs + CP, dots) for s-polarized and p-polarized laser light. As a result, differences between second-order NLO responses dependent on polarization have been noted. For p-p polarization, we observed a higher response, which means that generated second harmonic is polarized in a horizontal direction. Additionally, it was noticed that UV irradiation in the case of SHG had a positive effect—the NLO response is higher. The highest response was observed for the P2 sample. For the sample P1, which indicated the highest THG response, no SHG response was observed before UV irradiation, but, after that, a signal appeared. This increase is assigned to the isomerization of styrylquinoline units within the film, which locally generate structural modifications with changes in dipole moments of active chromophore groups. For the doped polymer with 10% of ZnO NPs of 100 nm, a considerable increase of the χ^(2)^ susceptibility is noticed if we compare with a polymer after the orientation technique (Table 2). SHG was not detected for P1 after corona poling and we can note the significant contribution of ZnO NPs to the nonlinear effects that stem from a photothermal heating, which can be considered as a result of energy transfer from a heated nanoparticle to the entire medium. The determined values for χ^(2)^ and χ^(3)^ are displayed graphically using the histogram in Figure 8 and Figure 9. Furthermore, the obtained results were compared with other organic guest-host samples L2ZnCl_2_ and L2AgNO_3_ with the poly(methyl methacrylate) PMMA matrix, which is already present in the literature [32], whose SHG and THG measurements were carried out using the same Maker fringe method. It can be noticed that the THG values obtained for investigated polymers are much higher than those already reported. Moreover, the SHG values for the P2 sample are much higher than complexes L2ZnCl_2_ and L2AgNO_3_. The reason of significant enhancement can be the strong acceptor moieties in the para position and electron-donor group (amino) and, as a result to the highest charge transfer in the polymer system, which has a strong impact on the NLO response of polymer.

## 4. Conclusions

In summary, nonlinear optical properties of polymers P1, P2, and P3 were performed by means of second and third harmonic generation using Nd:YAG laser with a fundamental wavelength of 1064 nm. SHG and THG measurements were carried out on two structurally different states of the films, induced by the irradiation at a 355-nm wavelength. Moreover, ZnO nanoparticles (NPs) (100 nm) were doped to the P1 polymer and the results of the nonlinear response were compared. Subsequently, based on the achieved SHG and THG responses, second and third order nonlinear susceptibilities, respectively, were determined based on theoretical comparative models. In the case of THG measurements, the highest signal of styrylquinoline polymer thin films was observed for the P1 sample. For samples irradiated with ultraviolet light and containing ZnO nanoparticles, the χ^(3)^ values do not differ significantly. Likewise, the polarization configuration had no effect on the obtained THG signals. For both the s-p and p-p polarization, no changes were observed in the THG signal intensity. On the other hand, in behalf of structural properties, the SHG studied had to be carried out using a corona poling technique. It was observed that UV irradiation and doping with ZnO nanoparticles enhanced the nonlinear SHG response in the case of sample P1. Moreover, for the P2 and P3 polymers, the UV irradiation enhanced the SHG signal by nearly twice in the case of sample P2, which resulted in this sample showing the best second order nonlinear properties. Marvelous properties of polymers P1, P2, and P3 presented in this manuscript show that they can be encouraging not only in a nonlinear optical application, but also in optoelectronics and photonics.

## Figures and Tables

**Figure 1 nanomaterials-11-00492-f001:**
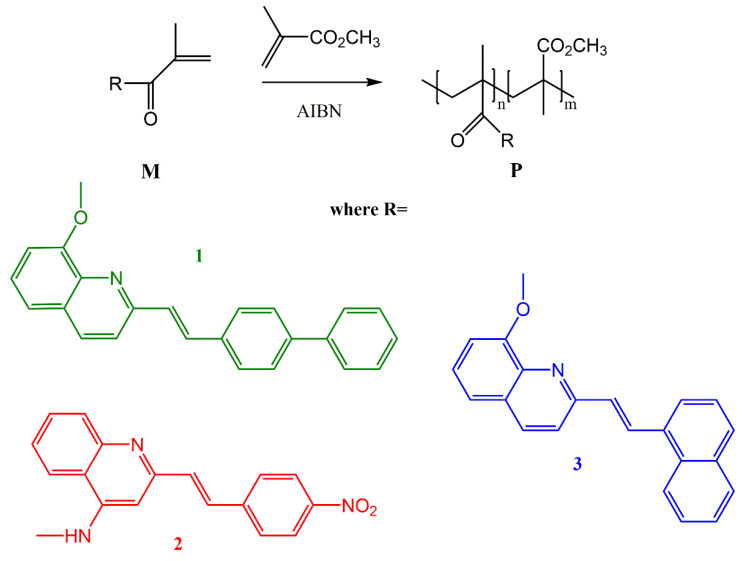
Synthesis of styrylquinoline copolymers by free radical polymerization P1, P2, and P3.

**Figure 2 nanomaterials-11-00492-f002:**
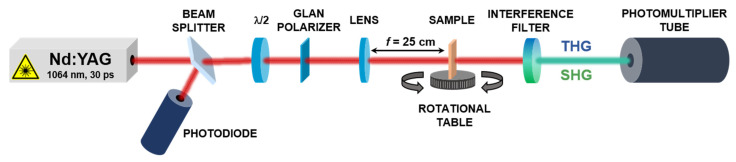
Second and third harmonic generation experimental setup. The following optical elements were used in the research: beam splitter for splitting the beam, one of which goes to the photodiode to detect the signal and possible noise, while the other passes through a half-wave plate, which changes the beam polarization. Then, after passing through Glan’s polarizer, according to which is possible to have less reflection loss, the sample passes to the lens at a distance of 25 cm from the sample. The interference filter positioned downstream of the sample is used to cut off nonessential signals and select the response to be studied. For SHG, it is a 532-nm filter. For THG, it is a 355-nm filter.

**Figure 3 nanomaterials-11-00492-f003:**
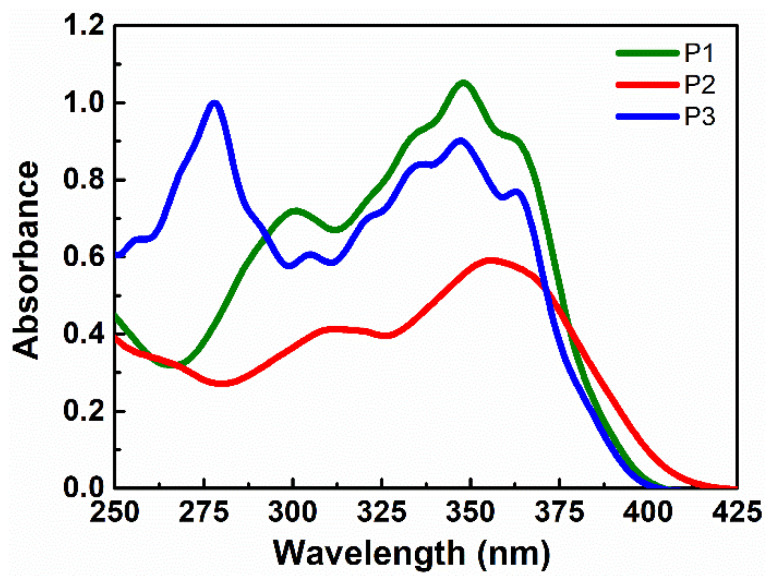
Absorption spectra of P1, P2, and P3 in tetrahydrofuran (THF) at room temperature.

**Figure 4 nanomaterials-11-00492-f004:**
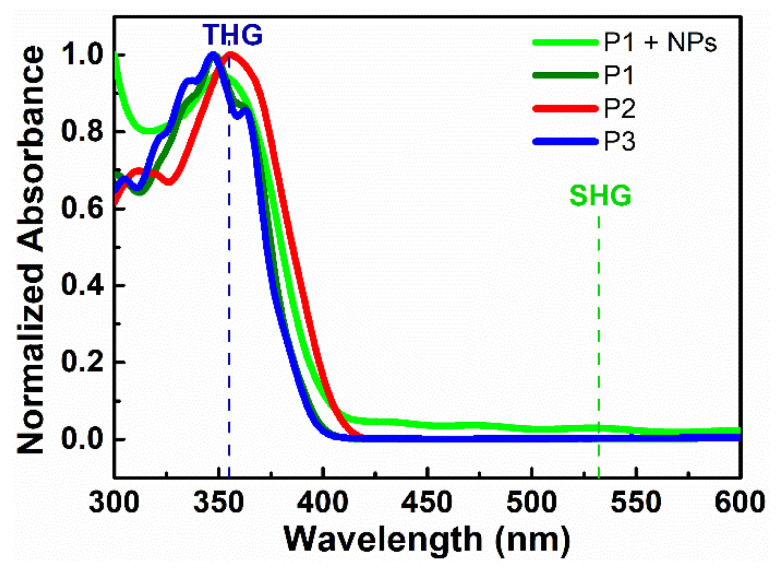
Normalized absorption spectra of studied polymeric thin films.

**Figure 5 nanomaterials-11-00492-f005:**
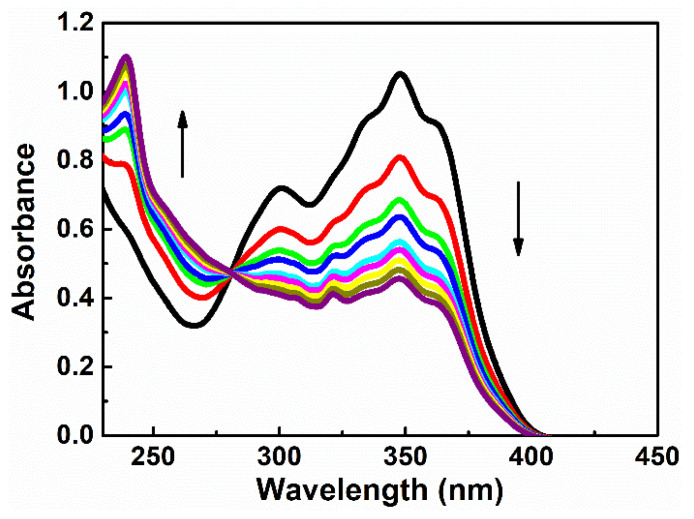
Changes in the absorption spectra of P1 in tetrahydrofuran (THF) before (1) and after 5 s period (2) 8 s (3), 12 s (3), 16 s (4), 17 s (5), 21 s (6), 25 s (7), 28 s (8), and 33 s (9) irradiation at a wavelength of 365 nm at room temperature.

**Figure 6 nanomaterials-11-00492-f006:**
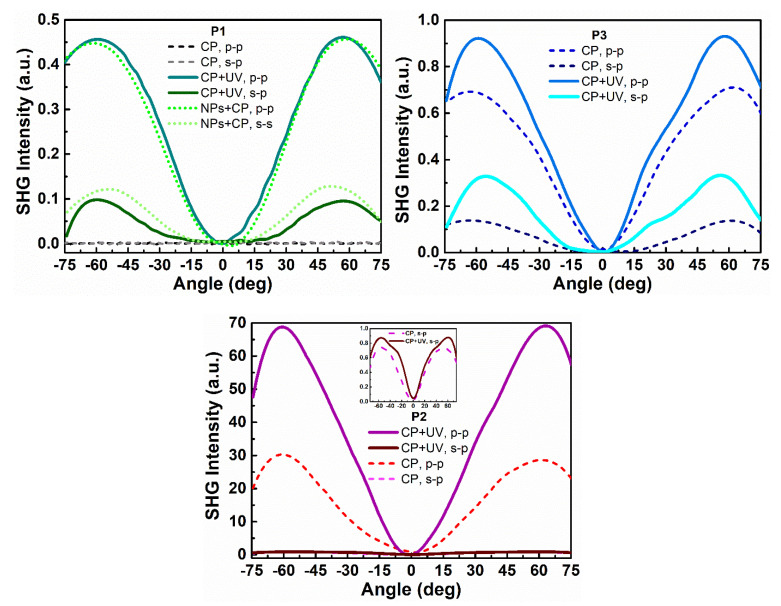
Second harmonic generation intensities as a function of rotation angle of polymers P1, P2, and P3, after corona poling (CP, dash line), UV–irradiation (UV, solid line) and with ZnO nanoparticles (NPs, dots) in an s–polarized and p–polarized laser beam.

**Figure 7 nanomaterials-11-00492-f007:**
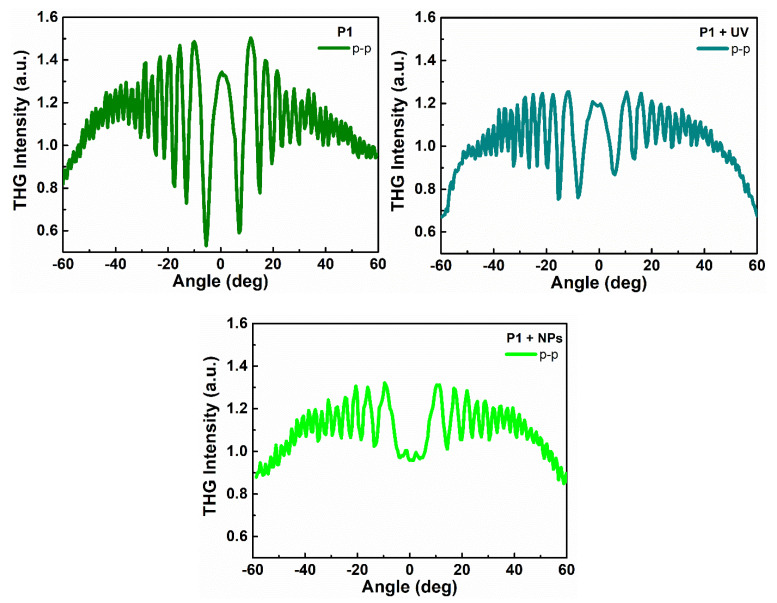
Third harmonic generation intensities as a function of rotation angle of polymer P1 after ultraviolet-irradiation (UV) and with ZnO nanoparticles (NPs) in a p-polarized laser beam.

**Figure 8 nanomaterials-11-00492-f008:**
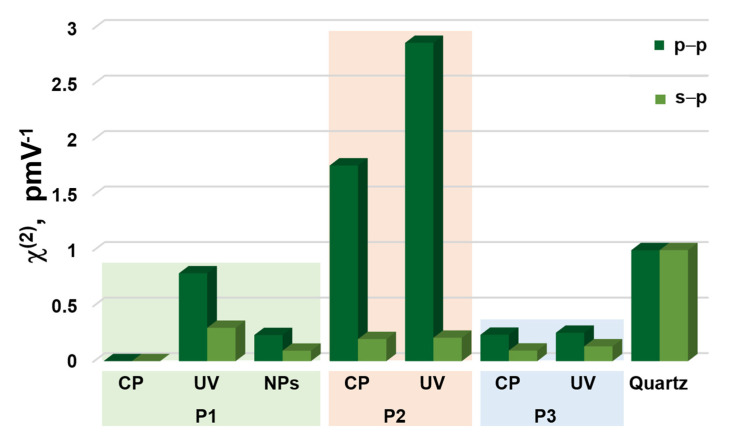
Histogram representing values of χ^(2)^ in s–p and p–p laser beam of polymer P1, P2, and P3 after corona poling (CP), UV–irradiation (UV), with ZnO nanoparticles (NPs), and comparison with reference material Y–cut quartz.

**Figure 9 nanomaterials-11-00492-f009:**
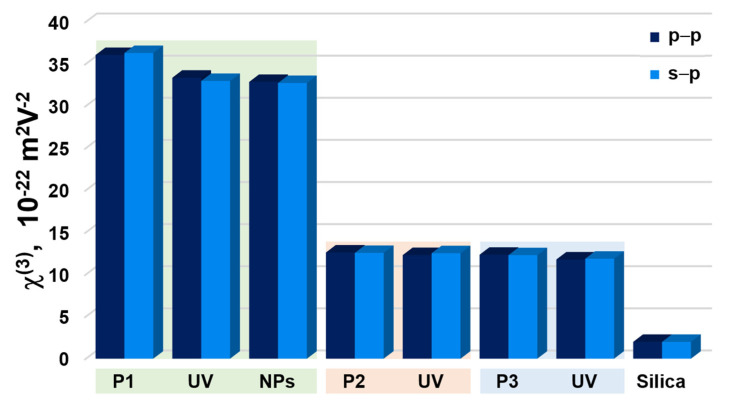
Histogram representing values of χ^(3)^ in s–p and p–p laser beam of polymer P1, P2, and P3 after UV–irradiation (UV), with ZnO nanoparticles (NPs), and comparison with reference material silica glass.

**Table 1 nanomaterials-11-00492-t001:** Characteristics of copolymers containing styrylquinoline units.

	P1	P2	P3
initial monomers mole ratio n/m	1:3		
ratio in copolymer n/m ^a^	1:2.6	1:3.2	1:2.8
Mw ^b^ [g/mol]	25,400	15,000	27,900
M_w_/M_n_ ^b^	1.84	2.0	1.9
T_g_ ^c^ [°C]	140	160	145

Determined: ^a^—by ^1^H NMR integration, ^b^—by GPC, ^c^—by DSC.

**Table 2 nanomaterials-11-00492-t002:** Values of thickness (d), absorption coefficient (α) at a wavelength of 355 nm, second (χ^(2)^) and third order (χ^(3)^) nonlinear susceptibilities in s-polarized and p-polarized laser beam after corona poling (CP), UV-irradiation (UV), and with ZnO nanoparticles (NPs) calculated for polymers P1, P2, and P3. Green arrows define increasing of χ^(2)^ and χ^(3)^ values, red arrows—decreasing of χ^(2)^ and χ^(3)^ values.

Sample	d [nm]	α_3__55_ × 10^3^[cm^−^^1^]	χ^(2)^ [pmV^−1^]		χ^(3)^ × 10^−22^ [m^2^ V^−2^]	
s-p	p-p		s-p	p-p	
P1	CP	368	4.89	-	-		(36.29 ± 1.1)	(36.05 ± 1.1)	
UV	(0.3004 ± 0.0188)	(0.7915 ± 0.0245)	↑	(32.97 ± 1.01)	(33.36 ± 1.02)	↓
NPs	1237	51.04	(0.0946 ± 0.0046)	(0.2355 ± 0.00396)	↓	(32.73 ± 0.538)	(32.86 ± 0.543)	↓
P2	CP	1350	10.18	(0.1992 ± 0.0029)	(1.763 ± 0.0213)		(12.56 ± 0.197)	(12.61 ± 0.198)	
UV	(0.211 ± 0.003)	(2.867 ± 0.0347)	↑	(12.52 ± 0.197)	(12.3 ± 0.195)	↓
P3	CP	1482	12.52	(0.0957 ± 0.0032)	(0.238 ± 0.00331)		(12.29 ± 0.192)	(12.34 ± 0.193)	
UV	(0.1332 ± 0.0027)	(0.2563 ± 0.00343)	↑	(11.87 ± 0.189)	(11.78 ± 0.188)	↓
L2ZnCl_2_ [32]	0.44	1.00		9.93	
L2AgNO_3_ [32]	0.32	0.96		7.92

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
