# Peer review of "Effect of UV-Irradiation and ZnO Nanoparticles on Nonlinear Optical Response of Specific Photochromic Polymers"

_nanomaterials, 2021, doi:10.3390/nano11020492_

Round 1

Reviewer 1 Report

The authors prepared ZnO nanoparticle doped polymer films on BK7 glass plates and characterized them. The preparation of the films is out of my area of expertise, so I will comment the characterization part.

  1. The first paragraph of the “Introduction” is hardly readable and contains grammatic errors too. This first paragraph needs to be rewritten and corrected absolutely. The other parts of the manuscript are much better, however, the entirely manuscript needs a careful reading. The senior members of the team should take the time and the responsibility to improve the overall stylistic and grammatic quality of the manuscript.
  2. Line 16 and 142: “pulse duration of 30 ps” instead of “repetition rate …”.
  3. Line 16: “dependence” instead of “depended”.
  4. The part between the lines 149-167 would fit better after line 129 and would be better to exchange it with the experimental setup.
  5. Line 187: “around” instead of “about”.

Author Response

Point 1: The first paragraph of the “Introduction” is hardly readable and contains grammatic errors too. This first paragraph needs to be rewritten and corrected absolutely. The other parts of the manuscript are much better, however, the entirely manuscript needs a careful reading. The senior members of the team should take the time and the responsibility to improve the overall stylistic and grammatic quality of the manuscript.

Response 1: Introduction has been significantly improved.

Point 2: Line 16 and 142: “pulse duration of 30 ps” instead of “repetition rate …”.

Response 2: Mistakes have been corrected.

Point 3: Line 16: “dependence” instead of “depended”.

Response 3: Mistake has been corrected.

Point 4: The part between the lines 149-167 would fit better after line 129 and would be better to exchange it with the experimental setup.

Response 4: Paragraphs have been exchanged.

Point 5: Line 187: around” instead of “about”.

Response 5: Mistake has been corrected.

Reviewer 2 Report

The article is dedicated to the synthesis of methacrylic styrylquinoline polymers and the study of NLO properties of polymeric layers based on them.  The article presents a number of original results regarding the influence of UV radiation, ZnO nanoparticles and radiation polarization on the second (SHG) and third (THG) harmonic generation.  It turned out that this effect is manifested in the change in the intensity of SNG only.  In this regard, the article can be published in the journal, but after the elimination of the following remarks:

 - there  is no justification for the choice of the structures of synthesized polymers, so this choice looks random;

 - there is no proper explanation for the sharp difference in the influence of the above factors on the properties of SHG and THG;

 - Table 2 shows the characteristics of SHG and THG for layers of L2ZnCl2 and L2AgNO3, but they are not discussed in the text of the article.

Author Response

Point 1: There  is no justification for the choice of the structures of synthesized polymers, so this choice looks random;

Response 1: We added to introduction (lines 36-41): “Among all types of polymers with chromophore fragments, polymers with styrylquinoline moiety have been chosen due to several advantages: styrylquinoline methacrylic polymer has chromophore with two functional groups, a central (ethylenic) double bond, and an endocyclic (quinoline) nitrogen atom, and as results photoisomerization of the ethylenic group allows switching between two isomeric states, trans(E) and cis (Z), excluding any side photochemical reactions; both isomeric states are thermally stable; protonation of the nitrogen atom allows shifting of the absorption spectrum; photoisomerization, and protonation process are reversible.”

Point 2: There is no proper explanation for the sharp difference in the influence of the above factors on the properties of SHG and THG;

Response 2: We added the missed explanation to the text (lines 247-250): “For the doped polymer with 10% of ZnO NPs 100 nm, a considerable increase of the χ(2) susceptibility is noticed if compare with polymer  after orientation technique. SHG was not detected for P1 after corona poling and we can be noted the significant contribution of ZnO NPs to the nonlinear effects can stem from a photothermal heating, which can be considered as a result of energy transfer from a heated nanoparticle to the entire medium.

Point 3: Table 2 shows the characteristics of SHG and THG for layers of L2ZnCl2 and L2AgNO3, but they are not discussed in the text of the article.

Response 3: We added in text (lines 251-257): “Furthermore, the obtained results were compared with other organic guest-host samples L2ZnCl2 and L2AgNO3 with PMMA matrix, already present in the literature [30], whose SHG and THG measurements were carried out using the same Maker fringe method. It can be noticed that the THG values obtained for investigated polymers are much higher than those already reported. Moreover, the SHG values for the P2 sample are much higher than complexes L2ZnCl2 and L2AgNO3. The reason of significant enhancement can be the strong acceptor moieties in the para position and electron-donor group (amino) and as result to the highest charge transfer in the polymer system, which has a strong impact on the NLO response of polymer

Reviewer 3 Report

  1. Line 16- instead of repetition rate 30 ps, it should be pulse duration 30 ps
  2. Line 142- repetition rate should be pulse duration and frequency of 10 Hz should be repetition rate
  3. Line 145-147 it is not clear how the authors made the p-p and s-p configuration ?
  4. Line181- there is typo TGH should be THG
  5. Line 189- there is typo adiabatic should be written together
  6. Figure 6- hard to follow some colors, authors can help to reader by using the different lines, like dash- dotted- etc
  7. It is not so clear that what authors would like to show in figure 6? Phase matching condition? They should discuss the figure 6 in the text better
  8. It is not so clear that corona poling is local effect or homogeneous in the film? If it Is not local, then it is hard to understand how it is possible to create macroscopic noncentrosymmetry. I think it is important to have a bit more clear explanation here
  9. Authors explained well why the UV radiation does effect on the SHG but I cannot see the explanation for ZnO nanoparticle doping? What is the mechanism behind it?

Author Response

Point 1: Line 16- instead of repetition rate 30 ps, it should be pulse duration 30 ps

Response 1: Mistake has been corrected.

Point 2: Line 142- repetition rate should be pulse duration and frequency of 10 Hz should be repetition rate

Response 2: Mistake has been corrected.

Point 3: Line 145-147 it is not clear how the authors made the p-p and s-p configuration ?

Response 3: We added in the text (lines 169-170): „The SHG and THG measurements were performed in two configurations by changing the angle on the half-wave plate (l/2), we are able to obtain s-polarized and p-polarized laser beam.”

Point 4: Line181- there is typo TGH should be THG

Response 4: Mistake has been corrected.

Point 5: Line 189- there is typo adiabatic should be written together

Response 5: Mistake has been corrected.

Point 6: Figure 6- hard to follow some colors, authors can help to reader by using the different lines, like dash- dotted- etc

Response 6: Graphs in figure 6 have been changed. We added dashed and dotted lines depending on the method used.

Point 7: It is not so clear that what authors would like to show in figure 6? Phase matching condition? They should discuss the figure 6 in the text better

Response 7: In Fig. 6 we would like to present the differences in the obtained response for individual samples depending on the input polarization, the corona poling technique used, UV irradiation and doping with nanoparticles. The appropriate technique is marked with different lines.

We added in the text (lines 237-241): „Figure 6 shows the second harmonic generation signal of P1, P2, P3 polymers after using the corona poling technique (CP, dashed line), then after irradiation with ultraviolet light (UV + CP, solid line) and a sample doped with ZnO nanoparticles (NPs + CP, dots) for s-polarized and p-polarized laser light. As a result, differences between second-order NLO responses dependent on polarization have been noted.

Point 8: It is not so clear that corona poling is local effect or homogeneous in the film? If it Is not local, then it is hard to understand how it is possible to create macroscopic noncentrosymmetry. I think it is important to have a bit more clear explanation here

Response 8: We added in text (line 236-237): “All studied thin films were poled at the same optimal conditions. The ordering is observed by the decrease of the thin film optical absorption at normal incidence after corona Poling [32].

[32] Page, R. H., Jurich,M. C., Reck, B., Sen, A., Twieg, R. J., Swalen, J. D., Bjorklund, G. C., & Wilson, C. G. (1990). J. Opt. Soc. Am. B, 7, 1239

Point 9: Authors explained well why the UV radiation does effect on the SHG but I cannot see the explanation for ZnO nanoparticle doping? What is the mechanism behind it?

Response 9: We added in the text (lines 247-250): “For the doped polymer with 10% of ZnO NPs 100 nm, a considerable increase of the χ(2) susceptibility is noticed if compare with polymer  after orientation technique. SHG was not detected for P1 after corona poling and we can be noted the significant contribution of ZnO NPs to the nonlinear effects can stem from a photothermal heating, which can be considered as a result of energy transfer from a heated nanoparticle to the entire medium.

Round 2

Reviewer 1 Report

The content and the style were essentially improved.

Reviewer 3 Report

The authors did enough improvement and clarification. I suggest that manuscript can be published in this form.